# Neutrophil Extracellular Traps Correlate with Tumor Necrosis and Size in Human Malignant Melanoma Metastases

**DOI:** 10.3390/biology12060822

**Published:** 2023-06-06

**Authors:** Lennard Marten Weide, Fiona Schedel, Carsten Weishaupt

**Affiliations:** Department of Dermatology, Skin Cancer Center, University Clinic Münster, Von-Esmarch-Str. 58, 48149 Münster, Germany; l_weid08@uni-muenster.de (L.M.W.); fiona.schedel@ukmuenster.de (F.S.)

**Keywords:** neutrophil extracellular traps, melanoma, neutrophils, cancer, necrosis, immune checkpoint inhibitors, metastasis

## Abstract

**Simple Summary:**

Neutrophil granulocytes are white blood cells that can release so-called neutrophil extracellular traps (NETs) which are composed of DNA and proteins. They help to fight pathogens such as bacteria but can also play a role in cancer. For human melanoma, which is due to its metastases the most lethal skin cancer, it is already known that neutrophils occur in metastases and can lead to a worse survival. Here, we investigate whether NETs can be found within melanoma metastases in order to better understand their role in the immune response to human melanoma and to assess their suitability as a potential therapeutic target. Therefore, we analyzed 81 metastases and found that one half was infiltrated with neutrophils and one third contained NETs. Further, we could see that NETs are predominantly found in large metastases and necrotic areas, i.e., areas with accumulated tumor cell death. NETs were present at all observed sites (skin, lymph node, liver and lung metastases). As many metastases contain NETs, our study delivers a basis to study NETs more intensively in the context of treatment response and the patient’s outcome in melanoma.

**Abstract:**

Neutrophil extracellular traps (NETs) are web-like structures released by neutrophils that kill invading microorganisms. However, NETs also promote tumor growth and impair the functionality of T-cells in cancer. Therefore, this study aimed at characterizing NET distribution within human melanoma metastases (n = 81 of 60 patients) by immunofluorescence staining for neutrophils (CD15) and NETs (H3Cit) in order to identify targets for NET-directed therapies. The results show that 49.3% of the metastases contained neutrophils (n = 40) and 30.8% (n = 25) contained NETs, 68% of them very densely infiltrated. A total of 75% of CD15-positive neutrophils and 96% of NET-containing metastases were necrotic while metastases without neutrophil infiltration were predominantly non-necrotic. A higher amount of NETs correlated significantly with greater tumor size. Consistently, all metastases with a cross-sectional area greater than 2.1 cm^2^ contained neutrophils. Analysis of metastasis from different sites revealed NETs to be present in skin, lymph node, lung and liver metastases. Taken together, our study was the first to observe NET infiltration in a larger cohort of human melanoma metastases. These results set the stage for further research regarding NET-directed therapies in metastatic melanoma.

## 1. Introduction

Malignant melanoma is the fifth most frequent tumor and shows an aggressive behavior once metastasized [1,2,3]. T-cell-based immunotherapies have revolutionized the treatment of advanced stages. Inhibition of the T-cell checkpoints PD-1 and CTLA-4 leads to a higher response rate compared to classical chemotherapies and even a long-term response over years is possible [4]. However, most patients do not experience this long-term response [5]. Thus, there is a high need to improve current immunotherapy protocols. In addition to T-cells, other immune cells play an important role in tumor defense. Tumor-associated neutrophil granulocytes (TANs) have emerged as a significant component of the tumor microenvironment (TME), exerting pro- or antitumor effects depending on disease stage, tumor type, and tissue [6]. In melanoma, TANs in primary tumors and metastases are associated with a worse prognosis [7,8].

An important aspect of TAN biology is the release of “neutrophil extracellular traps” (NETs) into the tissue or blood circulation. Under physiological circumstances, these “traps” serve to kill bacteria or fungi [9]. NETs consist of unfolded neutrophil chromatin fibers loaded with histones and many cytosolic and lysosomal neutrophil proteins including elastase, myeloperoxidase and metalloproteinase 9 [10,11,12].

NETs have been detected in tumor tissue and blood circulation of cancer patients with various tumors, suggesting a significant role in tumor immunology [13]. Of note, in a mouse model of colorectal cancer, disruption of NETs with DNAse I improves survival and leads to diminished tumor volume via enhanced CD8^+^ T-cell cytotoxicity, especially when combined with anti-PD-1 therapy [14]. In addition, in a pancreatic cancer model, where IL17A signaling enhances neutrophil recruitment and NET formation, inhibition of PAD4, an enzyme needed for NETosis that supports chromatin decondensation by citrullination of histone 3 [15], or IL17A blockade reverts CD8^+^ T-cell exclusion from the tumor mass and sensitizes to anti-PD-1 plus anti-CTLA-4 treatment [16]. Regarding melanoma, only one study to date has investigated the effect of PAD4 inhibitors. Interestingly, inhibition of NETosis skews neutrophils to an anti-tumor phenotype, preventing tumor growth in a murine melanoma model [17].

Hence, the question arises whether NETs represent a potential target additional to T-cell immune checkpoint inhibition in patients with metastatic melanoma. In our previous work, we demonstrated that NETs are present in human primary melanomas where they were associated with ulcerated areas and had tumor-inhibiting characteristics in vitro [18]. However, to address the potential use of NET inhibitors in advanced disease, the analysis of metastases is crucial. Therefore, this study aimed at analyzing neutrophil and NET distribution in human melanoma metastases.

## 2. Materials and Methods

### 2.1. Patients and Tissue Samples

Pathology registers of the University Clinic Münster’s Departments of Dermatology and Pathology were screened, and formalin fixed paraffin embedded (FFPE) tissue blocks of 81 metastases originating from 60 patients were finally included in this study (Appendix A). All metastases were examined by trained pathologists after excision. Patient data was retrospectively obtained from the internal clinical records.

### 2.2. Immunofluorescence Staining

FFPE blocks were cut into sections of 3 µm and deparaffinized. Heat-induced-epitope-retrieval (HIER) was performed with a pH6-citrate buffer for 20 min in a steam stove and 20 min at room temperature. The sections were then blocked for 30 min with 1% BSA and 10% normal goat serum in PBS. Subsequently, the primary antibodies followed by their corresponding secondary antibodies were applied each for 45 min at room temperature. For NET detection, we used a well-established anti-citrullinated histone 3 (H3Cit) antibody (#ab5103, Abcam, Cambridge, United Kingdom, 1:50) [17,19,20,21,22] followed by an AF-568-coupled secondary antibody (#A11036, Thermo Fisher Scientific, Waltham, MA, USA, 1:500). An anti-CD15 antibody (#301902, Biolegend, San Diego, CA, USA, 1:50) was used as neutrophil marker. Then, DAPI (#10236276001, Roche, Basel, Switzerland, 1:1800) and the second secondary antibody, coupled with AF-488, were added (#ab150121, Abcam, Cambridge, United Kingdom, 1:300). Between every step, the slides were washed and, in the end, mounted in Aquatex^®^ mounting medium (#108562, Merck Chemicals, Darmstadt, Germany). In every staining cycle, a primary melanoma known to contain neutrophils and NETs served as a positive control. As negative controls, sections were incubated with isotype antibodies matching the primary antibodies.

The staining for neutrophils was cross-validated with immunoperoxidase staining for CD15 (#CI063C01, DCS, Hamburg, Germany, 1:40), myeloperoxidase (MPO) (#A0398, Dako Agilent, Santa Clara, CA, USA, 1:20000) and neutrophil elastase (NE) (#M0752, Dako Agilent, Santa Clara, CA, 1:50) on selected metastases (Figure 1).

### 2.3. Histochemistry

H&E and immunoperoxidase staining for melanoma markers (i.e., Panmel, S100, MelanA, Sox10) were either obtained from the archives of the Department of Dermatology or Pathology or newly stained on serial sections of the metastases within the clinical routine.

### 2.4. Evaluation and Image Acquisition

Slides were observed under an Olympus BX63 fluorescence microscope (Olympus, Hamburg, Germany). Infiltration of neutrophils and NETs was assessed by scoring the signals for the green channel (CD15, “N”) and red channel (H3Cit, “NET”) separately as “without (0)”, “low (1)”, “medium (2)”, and “massive (3)” (see Table 1 and Appendix A for more details), also considering the isotype control. H3Cit signal limited to the intracellular space was not counted as NET-positive and was therefore categorized as “NET0”. Hence, only extruded NETs were considered. Every metastasis was scored twice, or, if infiltration detected, three times. Sections were independently evaluated by a second observer. Representative images of the regions with highest Infiltration were taken (Olympus XM10 camera, Hamburg, Germany).

To allocate NET-infiltrated regions to necrotic areas and melanoma tissue, images of H&E and peroxidase staining for melanoma markers (i.e., Panmel, MelanA, Sox10, S100) were simultaneously viewed. Areas were classified as “necrotic” when an eosinophilic cytoplasm, pyknotic or missing cell nuclei were detected. Necrosis limited to single cells was not categorized as “positive”. The categorization was cross-checked with the corresponding clinically obtained pathological records.

For size estimation, the cross-sectional area (CSA) of the metastases was measured on the same images (4× Olympus BX50 camera, Hamburg, Germany) using Fiji [23] (Figure 2A). Additionally, the values were validated by comparison with the maximal diameter known from the pathological records of 31 metastases (Appendix A).

### 2.5. Statistics

Statistical analysis was performed with GraphPad Prism (Version 9.4.0, San Diego, CA, USA). Dependence between two categorical variables was tested by Fisher’s exact test or chi-square test when appropriate. For analysis of continuous variables, the groups were tested for normal distribution by Shapiro–Wilk analysis and then tested by Kruskal–Wallis analysis (non-normal distribution) for comparisons including several subgroups. Direct comparison between two groups was performed by Mann–Whitney test (non-normal distribution) or unpaired *t*-test (normal distribution). For paired cross-validation of CSA and maximal metastatic diameter as mentioned in the clinical records, Wilcoxon matched-pairs signed rank test was used.

## 3. Results

A total of 49.3% (n = 40) of the 81 analyzed metastases contained neutrophils and NETs could be identified in 30.8% of the metastases (n = 25). Quantification of the neutrophils revealed 6 metastases with low (N1), 16 with medium (N2) and 18 with massive (N3) infiltration as defined in Table 1. Similar results were seen for NETs with infiltration being low (NET1) in 2, medium (NET2) in 6 and massive (NET3) in 17 of the infiltrated metastases. These results indicate that NETosis is a common phenomenon in human melanoma metastases and that if present, NETs cover large areas. To further characterize infiltrated and non-infiltrated metastases, we analyzed the correlations between presence of neutrophils and NETs with clinical-pathological characteristics of the metastases that may be of clinical interest, i.e., size, necrosis and metastatic site.

### 3.1. Greater Cross-Sectional-Area Correlates with Greater Intratumoral Infiltration of Neutrophils and NETs

As neutrophil infiltration and the number of NETs correlated with ulceration and necrosis in large primary melanoma, we wanted to analyze whether neutrophil and H3Cit infiltration correlates with the size of metastases. The size was assessed by measuring the cross-sectional-area (CSA) (Figure 2A) and had a median CSA of 68.5 mm^2^ among all tumors. The median CSA of neutrophil-infiltrated (148.7 mm^2^) and H3Cit-positive (151.0 mm^2^) metastases was significantly greater than those of metastases without infiltration with neutrophils (37.8 mm^2^) or H3Cit (58.04 mm^2^) (Mann–Whitney test; *p* < 0.0001 and *p* < 0.001) (Figure 2B). A comparison of CSAs of CD15 neutrophil-infiltrated metastases containing and not-containing NETs revealed no significant difference (Mann–Whitney test) (Figure 2C).

Interestingly, all metastases greater than 201.4 mm^2^ showed neutrophil infiltration (n = 13). On the other hand, the smallest metastasis with neutrophils had a CSA of 8 mm^2^. Moreover, no metastasis smaller than 25.34 mm^2^ contained NETs, meaning NET infiltration was limited to the largest 75% of metastases (Figure 2B).

Furthermore, the CSA size also correlated with the quantity of the neutrophil infiltrate (Kruskal–Wallis, N1 to N3, *p* = 0.0264 and N0 to N3, *p* < 0.0001). While low infiltrated (N1) metastases and not-infiltrated metastases (N0) had the same size, medium (N2) and massively (N3) infiltrated metastases were significantly larger (Mann–Whitney test, both *p* < 0.0001, compared to N0). Additionally, higher infiltration (N2, N3) was associated with greater size compared to low infiltration (N1) (Mann–Whitney test, N1-N2, *p* = 0.0487; N1-N3, *p* = 0.0034) (Figure 2B). Quantification of NETs revealed a greater size in medium (NET2) and massively (NET3) infiltrated metastases compared to metastases without NET infiltration (NET0) (Mann–Whitney test; *p* = 0.0495 and *p* = 0.0007). However, no difference was observed between low and massive infiltration.

In conclusion, greater size correlates significantly with increased neutrophil and NET infiltration with stronger effects seen for neutrophils compared to NETs.

### 3.2. Infiltration Is Strongly Associated with Necrosis

Since neutrophil and NET deposition is associated with ulcerated areas in primary melanoma [18], we analyzed whether there is a similar association with necrosis in melanoma metastases.

Necrosis was categorized as either positive (necrotic areas within metastasis) or negative (no necrosis/only single necrotic cells). A total of 30 (83.3%) out of 36 necrotic metastases contained neutrophils and 24 (66.6%) NETs (Figure 3A). Thus, neutrophils and NETs were significantly more abundant in necrotic metastases compared to non-necrotic metastases (n = 45) (Fisher’s exact test; *p* < 0.0001). Of the non-necrotic metastases, 10 metastases (22.2%) showed neutrophil infiltration and only in 1 (2.2%) cutaneous metastasis NETs (NET1) were found. Hence, 75% of neutrophils-containing and 96% of NET-containing metastases were necrotic.

Moreover, chi-square analysis revealed a strong association between necrosis and the different levels of infiltration for both neutrophils and NETs (both *p* < 0.0001). Based on the proportions of necrotic metastases within the different levels of infiltration, we attributed this association with necrosis primarily to the more highly with neutrophils and NETs-infiltrated metastases (Appendix A).

Within the infiltrated, necrotic metastases, the maximal density of neutrophil and NET infiltration was usually seen in the necrotic areas, but not limited to them (Figure 3B). However, larger scales of late-stage NETs, meaning intense H3Cit-positive signal alongside CD15 staining of neutrophils that already lost their physiological cell morphology, could only be seen in necrotic regions (Figure 3D). One cutaneous metastasis was ulcerated and showed massive infiltration with neutrophils and NETs in that area, consistent with our previous findings in ulcerated primary tumors. Of note, in some vessels in or adjacent to the tumors, increased H3Cit-positive neutrophils accumulated and partly extruded NETs into the vasculature (Figure 3E).

Metastasis size was shown to be highly associated with neutrophil and NET infiltration as well as necrosis as visualized for individual samples in Figure 4. The increased incidence of larger and necrotic metastases in the fields with higher infiltration underscores these observations.

We then compared the CSAs between infiltrated and non-infiltrated metastases in either necrotic or non-necrotic metastases. (Figure 3C). Here, a difference in CSAs of neutrophil-positive and negative metastases was revealed in both necrotic and non-necrotic metastases (Mann–Whitney, *p* = 0.026 and *p* = 0.0283). Again, for NETs this association could not be seen (Mann–Whitney, *p* = ns). A comparison of the infiltrate’s distribution within the different quartiles showed similar results (Appendix A).

Taken together, we see a strong association of the infiltrated metastases with necrosis, especially for NETs that are nearly limited to necrotic metastases. Corroborating this finding, the only ulcerated cutaneous metastasis present in this study showed massive neutrophil and NET infiltration, particularly in the ulcerated area.

### 3.3. Infiltration and Metastatic Site

The analyzed metastases originated from skin (n = 29), lymph nodes (n = 26), liver (n = 13) and the lungs (n = 11) (Figure 5C). For one metastasis, no clear assignment to either skin or lymph node could be made. For another one, it was not possible to distinguish a parotid from a lymph node metastasis, so that both have been excluded from this site dependent analysis. Both showed neutrophil as well as NET infiltration.

For the other metastases (n = 79), neutrophils and NETs were present in all observed sites. Within the metastases of each site most showed no neutrophil or NET infiltration, with liver metastases being the exception. In the liver, 76.9% of metastases (n = 10) contained neutrophils, which was significantly more than in lung metastases (27.2%, n = 3) (Fisher’s exact test, *p* = 0.0377), but also more than in skin (48.2%, n = 14) and lymph node metastases (42.3%, n = 11). However, chi-square analysis revealed no general dependence of neutrophil infiltration on the metastasis’ site (Figure 4A).

Concerning the infiltration with NETs, the highest relative amount of H3Cit-positive metastases was found in skin metastases (39.2%, n = 11) that also had the highest ratio of NETs within neutrophil-positive metastases (78.5%, n = 11/14), followed by liver metastases (38.4%, n = 5), lymph nodes (23.0%, n = 6) and lung metastases (9%, n = 1) (Figure 4A).

Looking at the sizes of metastases from the observed sites, liver metastases have been significantly greater than lung and skin metastases (Mann–Whitney test; *p* = 0.0175 and 0.0053). In addition, lymph node metastases have been larger than lung metastases (Mann–Whitney test; *p* = 0.0424). Due to this significant variance, we subsequently looked at the CSAs of infiltrated and not-infiltrated metastases after sorting for their metastatic site (Figure 5B). Throughout all metastatic sites, the neutrophil-infiltrated metastases showed a significantly greater CSA. For NETs, only infiltrated metastases of skin and lymph nodes were larger than not-infiltrated metastases. Again, size was primarily affecting the neutrophil infiltration. However, infiltration was not only associated with large size and large size did not necessarily mean that the metastasis is infiltrated with neutrophils or NETs.

Taken together, we could see no general association between metastatic sites and infiltration of the metastases with neutrophils and NETs. However, in direct comparison, liver metastases showed significantly more neutrophil-infiltrated metastases than lung metastases. The effects of size on metastatic infiltration as described above could generally be recapitulated in the site dependent analysis.

## 4. Discussion

Tumors modulate the host immune system resulting in inefficient immune responses [24]. Abrogation of NETs was shown to increase the capacity of T-cells to attack cancer cells [14,16]. Thus, melanoma, a pioneering tumor for T-cell-targeted immunotherapies, may represent a useful target for NET inhibitors in combination with checkpoint inhibitors. In order to check whether this is feasible, we investigated the occurrence of NETs in melanoma metastases. We found that about half of human melanoma metastases contained neutrophil granulocytes and about one-third of metastases showed NET infiltration. Indeed, the high amount of tumor-associated NETs in human melanoma metastases suggests them to be a potential therapeutic target for NET inhibitors. Our findings are in line with a previous study that observed neutrophil enrichments in melanoma tumors failing to respond to either anti-CTLA-4 or anti-PD-1 therapy [25]. Specifically referring to NETs in melanoma metastases, de Andrea et al. [26] and Munir et al. [17] detected NETs in n = 11 and n = 2 melanoma metastases, respectively. Furthermore, single-cell RNA sequencing of human brain metastases revealed genes associated with NETosis [27]. The present study confirmed these findings in a larger melanoma patient cohort including various metastatic sites. A limitation of our study is the retrospective design. It would be desirable to conduct prospective studies in the future to exclude confounding variables such as different pretreatment of melanoma metastases.

To understand the biology of infiltrated and non-infiltrated metastases, the clinical and pathological criteria were examined. NETs were present at all analyzed metastatic sites with liver metastases showing the highest infiltration of neutrophils. Further analysis revealed that liver metastases had a higher tumor volume than metastases of other sites. Considering all metastatic sites, the size of melanoma metastases significantly affected the recruitment of neutrophils and NETs. All metastases greater than 201.2 mm^2^, which corresponds to an estimated diameter of 14 × 17 mm, showed neutrophil infiltration. NETs were found exclusively in the largest 75% of metastases. We hypothesize that the correlation of NETs to the size of metastasis is due to the development of necrosis in hypoxic areas of large tumors.

In fact, the quantity of NETs was significantly associated with necrosis, similar to our previous results for primary melanoma, where infiltration was highly linked to ulcerated or necrotic areas [18]. Necrosis is an independent prognostic marker for worse survival in primary melanoma [28] and also in metastases where it occurs even more frequently [29]. The underlying mechanisms remain largely unknown, especially due to a lack of adequate necrosis models outside of whole organisms [30]. In our cohort, 36 of the 81 metastases contained necrotic areas, including 96% of the NET-containing metastases. NETs in or near necrotic areas also appear in other cancers and diseases [21,31]. Here, hypoxia [32] and DAMPs released by the necrotic areas, e.g., extracellular histone or tumor necrosis factor alpha (TNFa), can trigger neutrophils to undergo NETosis as it was shown for acute kidney injury (AKI) or ulcerative colitis [20,33].

Yee and Li recently proposed a tumor necrosis model in which neutrophils that have been recruited to the necrotic tumor site lead to increased necrosis by inducing ferroptosis in tumor cells [30,34]. They suggest a switch from immunogenic characteristics of initial necrosis to a more immunosuppressive type when the necrotic areas extend, supported by hypoxia and immunosuppressive metabolites that can reduce T-cell activity. NETs were mentioned to amplify this process by cutting off the blood supply through NET dependent thrombosis in tumor supplying vessels [30]. In line with this, our results in human melanoma metastases show that NETs can be found inside of vessel lumina close to the tumors (Figure 3E). Furthermore, NETs could potentially increase necrosis within the tumor directly as they were shown to induce necrosis in melanoma cells in vitro [18]. Inhibiting NETs might therefore not only enhance immunotherapy response rates by overcoming direct effects on T-cells as shown in a murine model by Kaltenmeier et al. [35], but a reduction in necrosis could further increase responsiveness through a less immunosuppressive tumor microenvironment. Indeed, it was shown by Nakazawa et al. in a mouse model for acute kidney injury that NET inhibition could decrease tissue necrosis [33]. First approaches to inhibit NETs in vivo have already been published and await further exploration [36,37].

## 5. Conclusions

Taken together, the results show that NETs are a frequently encountered phenomenon in malignant melanoma and thus merit evaluation as a potential therapeutic target. The finding that NETs are associated with necrosis highlights the importance of understanding the underlying mechanisms of NET formation in melanoma. Further studies are required to unravel the impact of tumor-associated NETs on the tumor microenvironment, tumor progression, patient prognosis and therapy response.

## Figures and Tables

**Figure 1 biology-12-00822-f001:**
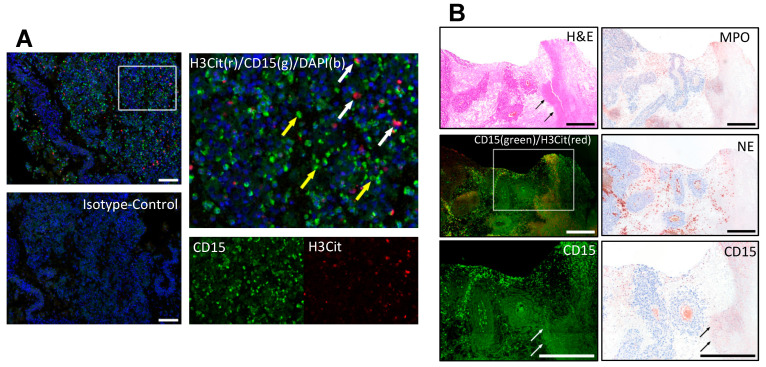
Immunofluorescence co-staining of neutrophils by CD15 (green) and NETs by citrullinated histone 3 (H3Cit, red). (**A**) Neutrophils (yellow arrows) and NETs (white arrows) inside a melanoma metastasis. Isotype controls were used to distinguish specific from background staining. (**B**) The neutrophil staining was further cross-validated using different neutrophil markers, myeloperoxidase (MPO), neutrophil elastase (NE) and CD15 in an immunoperoxidase staining to confirm the staining pattern especially in necrotic areas (black and white arrows) independent of autofluorescence. Scale bars = 100 µm (**A**) and 500 µm (**B**).

**Figure 2 biology-12-00822-f002:**
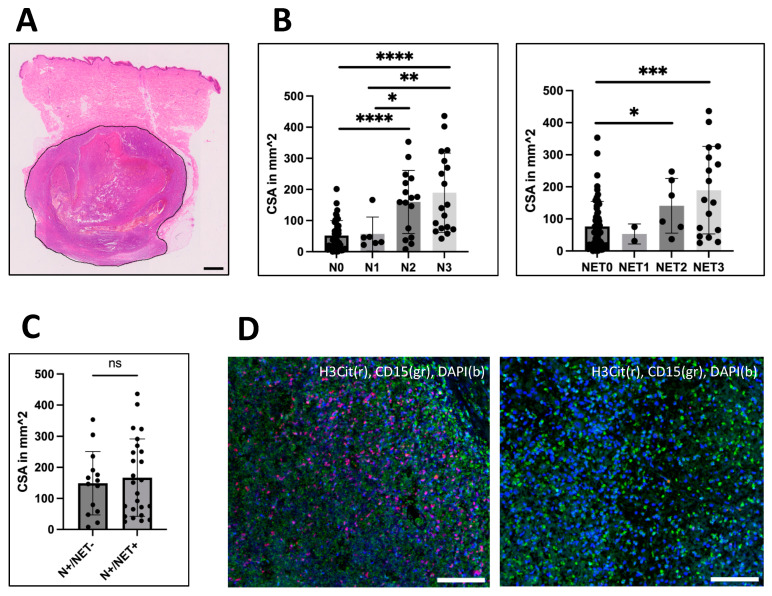
Analysis of correlation between size and infiltration. (**A**) The cross-sectional area (CSA) of the metastases was measured using H&E or peroxidase staining for melanoma-specific markers as here exemplarily shown (measured area surrounded in black). CSA varied from 0.04 mm^2^ up to 436 mm^2^ with a median size of 68.5 mm^2^. (**B**) CSA was correlated to the amount of neutrophil (left: N0-N3) and NET infiltration (right: NET0-NET3). For both, the median CSA was strongly distinct between medium and high infiltrated groups compared to not-infiltrated metastases. N1-infiltrated metastases were significantly smaller than N2 (*p* = 0.0487) and N3 (*p* = 0.0034)-infiltrated metastases, but no difference was seen between NET1 and NET2 or NET3-infiltrated metastases. (**C**) However, comparing CSA between metastases with CD15-positive neutrophils that contained NETs and those that did not, no association with size could be identified. (**D**) Left: Example of a massively neutrophil (N3) and NET (NET3)-infiltrated liver metastasis. Right: Lung metastasis with neutrophils (N3) but without NETs. Scale bars = 100 µm. Scale bar for (**A**) = 1 mm. * *p* < 0.05, ** *p* < 0.002, *** *p* < 0.001, **** *p* < 0.0001, ns = not significant. Error bars indicate mean ± SD.

**Figure 3 biology-12-00822-f003:**
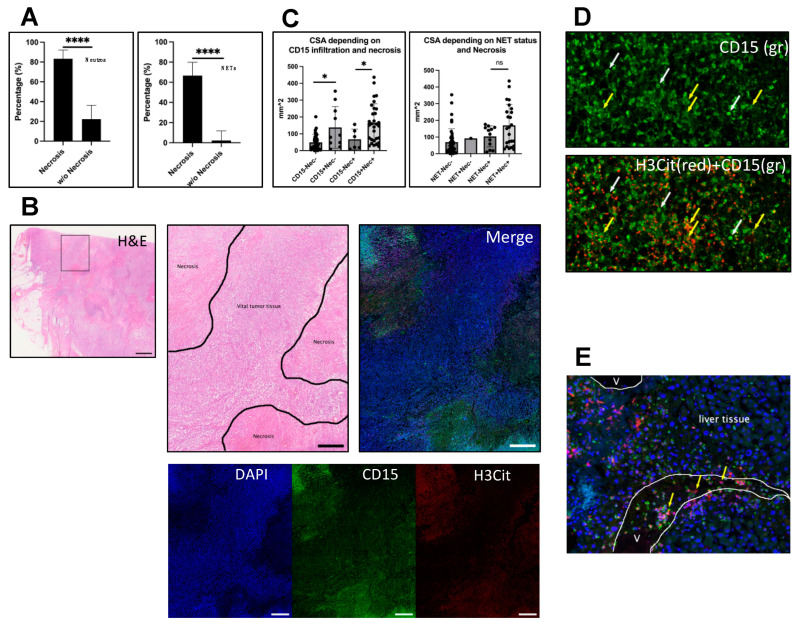
Infiltration is associated with necrosis. Necrosis was evaluated in corresponding H&E or IHC sections of the metastases and categorized as present or absent. (**A**) Left: Of the necrotic metastases (n = 36), 83.3% (n = 30) were infiltrated with neutrophils, whereas the proportion of neutrophil infiltration in non-necrotic metastasis (n = 45) was 22.2% (n = 10) (Fisher’s exact test, *p* < 0.0001). Right: A similar pattern could be seen for NETs. A total of 66.6% (n = 24) of necrotic metastases and only one non-necrotic (2.2%) metastasis contained NETs (Fisher’s exact test, *p* < 0.0001). Within the infiltrated metastases, the maximal neutrophil and NET density was predominantly found in necrotic areas. (**B**) Example of a necrotic cutaneous metastasis. Left: H&E overview. Box indicates magnified area shown in the other images presented in (**B**). Middle: H&E of necrotic areas (marked). Right: Corresponding image of an IF staining for DAPI (blue), CD15 (green) and H3Cit (red) shows neutrophils and NETs in the necrotic areas. Bottom: Single channels of the same image. (**C**) Analysis of CSAs of infiltrated and non-infiltrated metastases in necrotic and non-necrotic subgroups. CD15 neutrophil-positive metastases were significantly larger than neutrophil negative metastases in both necrotic and non-necrotic subgroups. Interestingly, for NETs, no significant difference regarding size occurred. (**D**) shows a necrotic region in a liver metastasis containing neutrophils with intact cell morphology (white arrows) and scattered CD15 signal (green), showing neutrophil debris after NETosis (yellow arrows) that could only be found in larger scales within necrotic areas. (**E**) shows a vessel near to a liver metastasis with accumulating neutrophils that form NETs (yellow arrows). V = vessel lumina. Scale bars: (**B**) overview: 1 mm; close-up: 500 µm. * *p* < 0.05, **** *p* < 0.0001, ns = not significant. The error bars in (**A**) indicate the 95% confidence interval. (**C**) presents the data with mean ± SD.

**Figure 4 biology-12-00822-f004:**
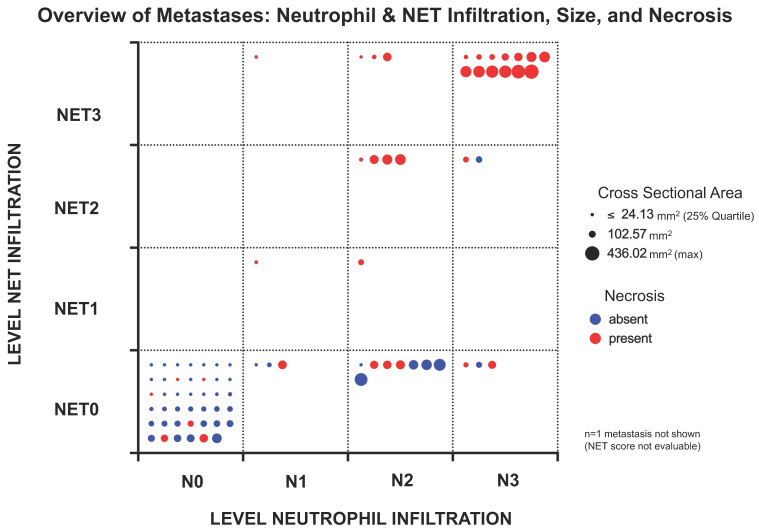
Overview of the analyzed metastases and their individual characteristics in terms of neutrophil and NET infiltration as well as size (as cross-sectional area (CSA)) and presence of necrosis. Each metastasis is represented by a dot whose size reflects the CSA. The smallest 25% of metastases are represented with the same size to ensure good visibility. Necrotic metastases are colored red, non-necrotic metastases are colored blue. Field position indicates the level of infiltration for neutrophils and NETs. Most metastases contained neither neutrophils nor NETs, which was usually associated with smaller CSA and absence of necrosis. Higher infiltration levels for both CD15-positive neutrophils and NETs were more prominent in the larger metastases, with NETs being almost limited to necrotic metastases. Neutrophil infiltration without NETs was also observed in some non-necrotic metastases.

**Figure 5 biology-12-00822-f005:**
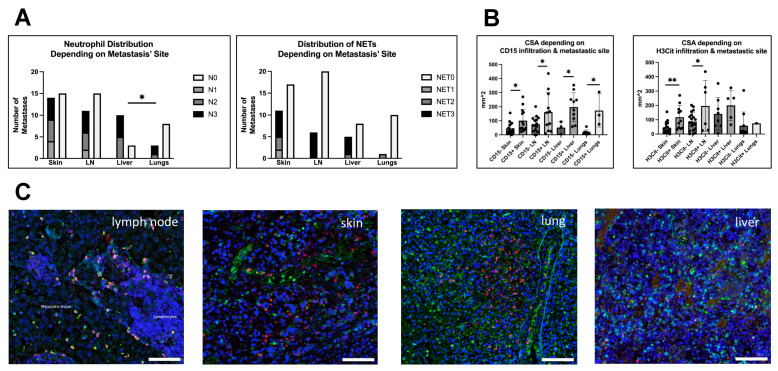
Neutrophil and NET infiltration could be found in metastases of all observed sites. (**A**) Left: Most liver metastases did show neutrophil infiltration, whereas in other sites, most metastases were not infiltrated by neutrophils, lung metastases even significantly less than liver metastases (Fisher’s exact test, *p* = 0.0377). Right: For NETs, liver metastases showed the highest infiltration ratio as well, but here infiltrated metastases build the minority throughout all sizes. (**B**) Because size was varying among metastatic sites, we compared CSAs of infiltrated and not-infiltrated metastases within the different locations. Left: Results for metastases infiltrated with neutrophils that continuously showed a greater CSA compared to not infiltrated metastases in all different metastatic sites. Right: For NETs, this was only true for skin and lymph node metastases. (**C**) shows immunofluorescence staining (for Neutrophils: CD15, green and NETs: H3Cit, red; DAPI, blue) of infiltrated metastases of different sites, i.e., (left to right) a lymph node metastasis (with tumor and lymphatic tissue), a cutaneous metastasis, a lung metastasis and a liver metastasis. Scale bars: 100 µm. * *p* < 0.05, ** *p* < 0.002. Error bars indicate mean ± SD.

**Table 1 biology-12-00822-t001:** Classification of infiltration levels for neutrophils and NETs.

Classification	Infiltration	Description
N0 or NET0	without	No or only sporadic neutrophils/NETs in entire metastasis
N1 or NET1	low	Several neutrophils/NETs spread throughout the metastasis
N2 or NET2	medium	Moderately distributed neutrophils/NETs throughout the metastasis
N3 or NET3	massive	Regions with very dense infiltration or distribution throughout the whole metastasis

## Data Availability

The data presented in this study is included in the main article and Appendix A. Additional material and raw data are available from the corresponding author upon reasonable request and limited by the protection of sensitive patient data.

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
