# Peer review of "Neutrophil Extracellular Traps Correlate with Tumor Necrosis and Size in Human Malignant Melanoma Metastases"

_biology, 2023, doi:10.3390/biology12060822_

Round 1
Reviewer 1 Report
Weide and coworkers investigated the presence of tumor-associated neutrophils (TAN) in melanoma-derived metastases located in various tissues derived from a larger number of patients. They show that the number of TAN and derived NETs correlated with the size of the metastases, irrespective of their tissue origin, and were apparent preferably in necrotic areas. In light of previous findings on the inhibitory role of intratumoral NETs for T cell responses, the authors conclude that TAN-inhibitory agents may be employed in combination with immune checkpoint inhibitors to improve anti-tumor T cell responses.
The study has been well-conducted and the results are presented in a concise manner.
This study aimed to elucidate at which extent tumor-associated neutrophils (TAN) and TAN-derived NETs were apparent in melanoma metastases in different organs of patients. This issue is important with regard to the importance of TAN-inhibiting drugs for the treatment of metastases.
The topic is interesting insofar as it sheds light on the prevalance of TAN as a component of the tumor microenvironment (TME) within melanoma metastases. In light of the role of TAN-derived NETs peviously reported to promote tumor growth by inhibiting T effector cells the study underscores to develop/apply TAN-inhibiting and reprograming agents, respectively, for the treatment of metastases as an approach complementary to other regimes like immune checkpoint inhibitors and chemotherapeutics, respectively.
The manuscript focused to delineate the prevalance of TAN and TAN-derived NETs in melanoma metastases in different organs from a larger patient cohort. The results indicate a strong correlation between the size of metastases and TAN/TAN-derived NETS as well as necrotic regions. Thereby the study emphasizes the importance for the development of TAN-targeting drugs for treatment of metastases.
The methods applied are suitable to address the issue. One could envision to costain additional components of the TME to delineate potential interaction in future studies.
The conclusions are in line with the proposed hypothesis and are based on the results of the study.
The references are appropriate as they cover the phenotype of neutrophils/TAN and NETosis as a primary mean for pathogen killing, but also inhibiting anti-tumor T cells and comprise TAN-inhibiting strategies.
The figures are well composed since they show in each case an example of immuno-stained tissue and derived quantification results.
Reviewer 2 Report
The authors aimed to investigate the roles of neutrophil extracellular traps in metastatic melanoma lesions. There are some issues or suggestions below:
Introduction / Discussion:
line 64, 327: Provide a more comprehensive background on the current findings in assessing the NET in melanoma
Methods:
line 96: Describe the reason of NET marker selection and justification for using this marker in assessing the NET
line 123: Justify the choice of statistical methods used to analyze the data. Reference(s) with a similar approach and the sample size should be cited.
Results:
line 162: Consider mapping each sample with levels of its neutrophil/NET infiltration and CSA
line 225: Consider presenting a classification diagram, Venn diagram or alternatives illustrating the distribution and classification of infiltration levels (i.e., neutrophil and NET infiltration) across both metastases and non-metastases samples, as well as neurosis lesions if available
line 260: Figure 4A, bars for neutrophil and NET should be presented with N1-N3 or NET1-3
Discussion:
line 344: Elaborate the strengths and limitations of your study, including any potential sources of bias or confounding factors, if available
Format:
line 188: consider formatting the numbered subtitle; it should be 3.2.
line 226: consider formatting the numbered subtitle; it should be 3.3.
Minor editing of the language or format is required.
